# Spinetoram-Induced Potential Neurotoxicity through Autophagy Mediated by Mitochondrial Damage

**DOI:** 10.3390/molecules29010253

**Published:** 2024-01-03

**Authors:** Fan Chen, Jin Lu, Meng Li, Junwu Yang, Wenping Xu, Xufeng Jiang, Yang Zhang

**Affiliations:** 1Shanghai Key Laboratory of Chemical Biology, School of Pharmacy, East China University of Science and Technology, Shanghai 200237, China; 18915143783@163.com (F.C.); amy.lu@qwwz.com (J.L.); 17864237231@163.com (M.L.); xuwp@ecust.edu.cn (W.X.); 2Frog Prince (Fujian) Baby&Child Care Product Co., Ltd., Zhangzhou 363000, China; yangjunw@qwwz.com; 3Ugel Cosmetics PTE Ltd., Singapore 349561, Singapore

**Keywords:** spinetoram, neurotoxicity, autophagy, human SH-SY5Y cell line, zebrafish, risk assessment

## Abstract

Spinetoram is an important semi-synthetic insecticide extensively applied in agriculture. It is neurotoxic to insects, primarily by acting on acetylcholine receptors (nAChRs). However, few studies have examined the neurotoxicity of spinetoram in human beings. In this study, various concentrations (5, 10, 15, and 20 μM) of spinetoram were employed to expose SH-SY5Y cells in order to study the neurotoxic effects of spinetoram. The results showed that spinetoram exposure markedly inhibited cell viability and induced oxidative stress. It also induced mitochondrial membrane potential collapse (ΔΨm), and then caused a massive opening of the mitochondrial permeability transition pore (mPTP), a decrease in ATP synthesis, and Ca^2+^ overloading. Furthermore, spinetoram exposure induced cellular autophagy, as evidenced by the formation of autophagosomes, the conversion of LC3-I into LC3-II, down-regulation of p62, and up-regulation of beclin-1. In addition, we observed that p-mTOR expression decreased, while p-AMPK expression increased when exposed to spinetoram, indicating spinetoram triggered AMPK/mTOR-mediated autophagy. Complementarily, the effect of spinetoram on neurobehavior was studied using the zebrafish model. After being exposed to different concentrations (5, 10, and 20 μg/mL) of spinetoram, zebrafish showed neurobehavioral irregularities, such as reduced frequency of tail swings and spontaneous movements. Similarly, autophagy was also observed in zebrafish. In conclusion, spinetoram exposure produced potential neurotoxicity through autophagy mediated by mitochondrial damage. The experimental data and results of the neurotoxicity study of spinetoram provided above are intended to serve as reference for its safety assessment.

## 1. Introduction

Pesticides are important in agricultural production; on the one hand, they can promote agricultural production; on the other hand, they may also cause potential health risks to human beings [1]. Spinetoram, an important semi-synthetic insecticide, consists of two primary active ingredients: spinetoram-J and spinetoram-L in an approximate 3:1 ratio [2]. It belongs to the spinosad family isolated from actinomycete *Sacharopolyspora spinosa* and has insecticidal activity against the lepidopteran, diptera, and thysanoptera pests [3]. Spinetoram is also useful in stopping pests from infesting. Some studies show that spinetoram is roughly twice as poisonous to houseflies as polymyxin [4]. Sheele [5] used spinetoram to remove human blood-sucking bedbugs, which are ectoparasites that thrive in temperate climates. Additionally, spinetoram effectively controls parasites in livestock and pets with low toxicity and easy biodegradability, minimizing any potential harm to animals. Multiple studies have shown the effectiveness of spinetoram in managing pet parasites. Franc et al. [6] used spinetoram tablets to treat flea infestation in cats. Paarlberg et al. [7] compared various drugs for treating cat fleas and discovered that spinetoram was highly effective with an efficacy rate of over 96.0% in a 0–37-day period. Additionally, spinetoram exhibited stronger insecticidal properties than fipronil and showed no adverse reactions.

Spinetoram controls insects primarily by acting on the nicotinic acetylcholine receptor (nAChR) to disrupt nerve signal transmission [8]. Spinetoram is used worldwide due to its high biological activity [9] and high degradation rate with a half-life of 0.35–4.25 days [10]. An increasing number of studies have shown the toxic effects of spinetoram on non-target organisms, including developmental toxicity, immunotoxicity [11], and hepatotoxicity [12,13]. However, scant research has revealed the neurotoxicity of spinetoram on human beings.

Neurotoxicity reflects damage to the structure or function of the central or peripheral nervous system [14]. The nervous system controls most of the body’s activities. It is sensitive to external perturbations, and minor structural, physiological, and biochemical damage to the nervous system can be detrimental to human health [15]. The neurotoxicity of spinetoram to humans is worth studying. Having appropriate in vitro and in vivo systems contributes to the study of neurotoxicity. SH-SY5Y cells belong to chromosomally stable human neuroblastoma cell line. Compared with other neuroblastoma cell lines, such as rat B35, mouse Neuro-2A cells, and rat PC12 cells, SH-SY5Y cells more closely resemble human cells [16]. SH-SY5Y cells possess many properties of dopaminergic neurons, including the expression of dopamine-β-hydroxylase and tyrosine hydroxylase, as well as dopamine transporter activity, and they have been widely used as a dopaminergic neuronal cell model for neurotoxicity studies [17,18]. In vivo models, zebrafish share great physiological and genetic similarities with humans, and have the advantage of having transparent embryos, short developmental cycles, and easy observation of neuromotor behavior [19]. Zebrafish has been proposed as an in vivo model to detect specific neurological alterations/pathologies during early developmental stages [20]. Therefore, a combination of SH-SY5Y cell and the zebrafish model was used in this study.

Autophagy is the process of cellular conserved degradation to remove damaged components and organelles. Cellular damage is largely under the control of the highly regulated process of autophagy. Autophagy is a key link that causes cell death and an important mechanism to maintain the balance of biological processes [21]. The deregulation of autophagy is a hallmark of many diseases [22]. Mitochondria are the cellular powerhouses that generate energy through the process of oxidative phosphorylation and play critical roles in cellular autophagy [23]. Thus, spinetoram-induced neurotoxicity can be explored from the perspective of the autophagy of the SH-SY5Y cell. In addition, in vertebrates, all behavior is achieved through neural control. The alterations in neural function are most evident in the behavioral “phenotype” (observable manifestation) [20]. Thus, it is necessary to examine the effects of spinetoram on neurobehavior during zebrafish development as a complement to neurotoxicity studies.

In this study, we investigated the neurotoxicity of spinetoram in humans based on SH-SY5Y cell and zebrafish models. First, SH-SY5Y cells were exposed to spinetoram to examine cell viability, autophagy, DNA damage and mitochondrial injury. Then, zebrafish embryos were exposed to spinetoram to observe their neuromotor behavior, including tail swings and spontaneous movements. Taken together, this information enriches the understanding of spinetoram-induced neurotoxicity.

## 2. Results

### 2.1. Spinetoram Exposure Inhibited SH-SY5Y Cell Viability

MTT results showed that the cell survival decreased significantly with increasing spinetoram concentrations (Figure 1). LC_50_ values were calculated to be 12.6 μM (24 h) and 3.34 μM (48 h). The result indicated that spinetoram exposure inhibited SH-SY5Y cell viability.

### 2.2. Spinetoram Induced Mitochondrial Dysfunction in SH-SY5Y Cells

The mitochondrial membrane potential (ΔΨm) is a global indicator of mitochondrial function and can be evaluated by measuring the relative fluorescence of Rhodamine 123. As shown in Figure 2A, the fluorescence intensity presented a general decline after exposure to spinetoram, which suggested spinetoram caused the reduction of ΔΨm in SH-SY5Y cells. The mPTP is a group of protein complexes located between the inner and outer mitochondrial membranes that play an important role in cell survival. The mPTP can be detected using the fluorescent probe calcein-AM and CoCl_2_. Figure 2B shows that the fluorescence intensity was strong following calcein staining without CoCl_2_, indicating equivalent intracellular calcein-AM loading. When CoCl_2_ treatment was applied, the fluorescence intensity decreased, indicating that CoCl_2_ quenched the green fluorescence of Calcein. Notably, the fluorescence intensity was significantly reduced when exposed to spinetoram compared to the control group, indicating the mPTP opening after spinetoram treatment. ATP is an energy substance produced by mitochondria, and we found that the ATP level in the cells was reduced after spinetoram exposure (Figure 2D). Ca^2+^ is a signaling messenger that can be measured with the fluorescent probe Fluo-3 AM. It was found that the fluorescence intensity increased obviously with the increase of the spinetoram concentration (Figure 2E,F).

### 2.3. Spinetoram Induced Oxidative Damage in SH-SY5Y Cells

Oxidative stress can be measured by ROS levels and expression of key enzymes (SOD, CAT, GSH-Px, MDA). ROS is an important marker of oxidative damage, and it was found that the ROS level significantly increased after EMB exposure compared to the control group (Figure 3A,B). Furthermore, the activity of SOD was remarkably reduced, while the activity of CAT, GSH-Px, and MDA were remarkably elevated after exposure to spinetoram (Figure 3C–F). The results indicated that spinetoram induced oxidative damage in cells.

### 2.4. Spinetoram Induced Autophagy through AMPK/mTOR Signaling Pathways in SH-SY5Y Cells

Monodansylcadaverine (MDC) dye can label autophagosomes. As shown in Figure 4A,B, the intensity of green fluorescence increased with increasing spinetoram exposure concentration, suggesting spinetoram induced the formation of autophagosomes. Beclin-1, p62, LC3-I, and LC3-II are autophagy-related proteins involved in autophagy flux. In the western blot experiment results, the protein expression ratio of LC3-II/I and beclin-1 were up-regulated, while that of p62 was down-regulated simultaneously after spinetoram treatment (Figure 4C,D). The above results indicated that spinetoram was sufficient to induce autophagy in cells. In addition, adenosine-monophosphate activated-protein kinase (AMPK) and mammalian target of rapamycin (mTOR) are major regulators of autophagy. We found that the phosphorylation level of AMPK drastically increased and that of mTOR was significantly suppressed after exposure to spinetoram (Figure 4E,F). The results suggested spinetoram induced autophagy through AMPK/mTOR signaling pathways in cells.

### 2.5. Spinetoram Altered Neuromotor Behavior in Zebrafish

The neuromotor behaviors of zebrafish during development affected by spinetoram were investigated. The results found that when zebrafish were exposed to spinetoram for 120 hpf, the number of tail swings in embryos gradually decreased (Figure 5A,B). It was also found that when a zebrafish was exposed to spinetoram up to 120 hpf, its voluntary movements gradually decreased (Figure 5C). Thus, spinetoram could damage the neuromotor behavior in zebrafish.

### 2.6. Spinetoram Induced Autophagy in Zebrafish

LC3 is a marker protein of autophagy. The autophagy in the zebrafish was assessed through LC3 immunofluorescence approach. As shown in Figure 5D,E, the green fluorescence spots on the head of zebrafish were dramatically enhanced after exposure to spinetoram for 120 hpf, suggesting that spinetoram exposure led to autophagy in zebrafish.

## 3. Discussion

In this study, we investigated the neurotoxicity in response to exposure to spinetoram based on the human SH-SY5Y cell line and zebrafish model. Firstly, the study investigated the effects of spinetoram on the SH-SY5Y cell. The findings revealed a significant decrease in cell survival after a 24 h exposure to spinetoram, with an LC_50_ value of 12.6 μM. Notably, in previous studies, the 24 h LC_50_ of polymyxin against HepG2 cell was 10.73 μM, and the mechanism of hepatocyte toxicity has been interpreted as programmed cell death [12,13]. However, the mechanism of toxicity in neuronal cells has not yet been investigated. Therefore, we further investigated the underlying mechanism of spinetoram-induced cytotoxicity in SH-SY5Y cells.

Mitochondria are the “energy factories” of our body, providing fuel for normal cellular function [23]. Mitochondrial membrane potential (ΔΨm) can reflect the integrity of mitochondrial function. Mitochondrial permeability transition pore (mPTP) is a non-specific channel and the prolonged opening of the mPTP permits rapid passage of ions and macromolecules, which may lead to cell death. ATP is the material that directly supplies energy to cells, and Ca^2+^ is the signaling messenger, both of which are essential for maintaining life activities [24]. In this study, we found that spinetoram exposure induced the collapse of ΔΨm. Subsequently, spinetoram caused the opening of mPTP inside the mitochondrial membrane, which induced the excessive cellular Ca^2+^ uptake and inhibited the adenosine triphosphate (ATP) synthesis. The results suggested that cell exposure to spinetoram produced an impairment of mitochondrial function.

Oxidative stress reflects an imbalance between the oxidative system and the antioxidant system, which ultimately caused excessive oxidative damage to living organisms [25]. The reactive oxygen species (ROS) produced by the mitochondrial respiratory chain represents the key inducer of oxidative stress [26]. In addition, superoxide dismutase (SOD), glutathione peroxidase (GPX), catalase (CAT), and malondialdehyde (MDA) are important antioxidant enzymes that are frequently considered as the markers of oxidative stress [25,27]. In our study, ROS level was dramatically increased in a dose-dependent manner under spinetoram exposure, which indicated spinetoram disturbed the balance between the ROS and antioxidant defenses. Furthermore, the level of SOD in spinetoram-treated SH-SY5Y cells was obviously reduced, while the level of GPX, CAT, and MDA were obviously increased, suggesting the occurrence of oxidative stress.

It is well known that oxidative stress could activate the process of autophagy [26]. Autophagy is an important biological process, and the damage of autophagy is closely related to mitochondrial dysfunction [28]. Autophagy is characterized by the formation of double-membrane vesicles called autophagosomes [29]. Microtubule-associated protein 1 light chain 3 alpha (LC3) is considered as a marker of autophagy [30]. In our experiments, we observed the formation of autophagic vacuoles and the conversion of LC3-I into LC3-II after spinetoram treatment. Furthermore, p62, as an autophagy substrate, plays an important role in the process of autophagy degradation [31]. Beclin-1 is a significant autophagy effector which is involved in the regulation of autophagy [32]. We found that the expression of p62 was remarkably down-regulated, while the expression of beclin-1 was remarkably up-regulated when exposed to spinetoram. Those findings supported the results that spinetoram could facilitate autophagy in SH-SY5Y cells.

We further explored the relevant signaling pathways of autophagy in spinetoram-treated SH-SY5Y cells. Generally, autophagy is regulated by two main kinases containing the AMP-activated protein kinase (AMPK) and mechanistic target of rapamycin (mTOR). Among them, AMPK is phosphorylated and activated in a metabolically adverse environment, while mTOR is phosphorylated and activated in a metabolically favorable environment. Our results showed that spinetoram remarkably elevated the phosphorylation level of AMPK and diminished the phosphorylation level of mTOR [33]. Based on this result, we proposed that spinetoram-mediated AMPK/mTOR signaling pathway resulted in SH-SY5Y cells autophagy.

In addition to studying the toxicity of spinetoram exposure on SH-SY5Y cells, the effect of spinetoram on the neurobehavior of zebrafish was also investigated. The spontaneous movements of zebrafish represent the biological function of the central nervous system [34]. In this study, we evaluated the effect of exposure to environmentally relevant doses of spinetoram on spontaneous movements in zebrafish. It was found that zebrafish embryos exposed to spinetoram for 16 hpf showed a significant reduction in the number of tail swings. Spontaneous locomotion rates in zebrafish were also significantly reduced after exposure to spinetoram at 120 hpf, particularly at 20 µg/mL (about 27 µM), which is less than half of that observed in the control group. In an abamectin insecticide study [35], a similar result was observed at approximately 0.5 µM.

The above results suggested that spinetoram disrupted neurobehavioral functions. On the other hand, LC3 immunofluorescence results demonstrated that spinetoram exposure, particularly at high concentrations of 10 μg/mL and 20 μg/mL, similar to those in the cell assay, induced autophagy in the head of a zebrafish larva. This is consistent with our experimental findings in SH-SY5Y cells. Combined with the findings of the cellular experiments, the abnormalities in zebrafish neurobehavior may be caused by its neuronal autophagy triggered through the AMPK/mTOR-signaling pathway.

In conclusion, the aim of this study was to investigate the neurotoxicity of spinetoram in humans using SH-SY5Y cells and the zebrafish model. The results showed that spinetoram exposure induced cytotoxicity in SH-SY5Y cells and triggered AMPK/mTOR-mediated autophagy, which may be associated with mitochondrial damage and ROS-mediated oxidative damage. In addition, we also observed that spinetoram exposure can induce neurobehavioral abnormalities and autophagy in zebrafish. Our results provide new insights into the neurotoxicity effects of spinetoram on human beings and provide some guidance for the safety risk assessment of spinetoram.

## 4. Materials and Methods

### 4.1. Chemicals and Reagents

Spinetoram (CAS No: 187166-40-1, purity > 98%) was purchased from Sigma-Aldrich (St. Louis, MO, USA). Dulbecco’s modified Eagle’s medium (DMEM) and penicillin–streptomycin were obtained from Hyclone (Logan, UT, USA). Fetal bovine serum (FBS) was obtained from Gibco (Norristown, PA, USA). Phosphate buffered saline (PBS) was obtained from Servicebio (Wuhan, China).

The malondialdehyde (MDA) content assay kit (D799762-0100), reduced glutathione (GSH) content assay kit (D799614-0100), catalase (CAT) activity assay kit (D799598-9100), and superoxide dismutase (SOD) activity assay kit (D799594-0100) were bought from Sangon Biotech (Beijing, China). The LC3 antibody, Beclin1 antibody, p62 antibody, p-mTOR antibody, p-AMPK antibody, GAPDH antibody, and γH2AX antibody were purchased from Cell Signaling Technology (Danvers, MA, USA). The Alexa fluor 488-conjugated antibody were purchased from Sangon Biotech (Shanghai, China).

### 4.2. Cell Culture and Treatment

Human neuroblastoma cell line SH-SY5Y cells (ATCC, Manassas, VA, USA) were grown in a DMEM medium containing 10% FBS and 1% penicillin–streptomycin and cultured in a sterile incubator at 37 °C with 5% CO_2_. The cells were cultured for no more than 20 passages. A control group was established with different concentrations (5, 10, 15, and 20 μM) of spinetoram-treated groups. In the control group, cells were exposed to 0.1% DMSO, and in the spinetoram-treated groups, spinetoram was dissolved in DMSO to form a stock solution (10 mM) and diluted with a medium to form the final concentration.

### 4.3. Cell Viability Assay

The cell viability was assessed via the MTT method. Briefly, SH-SY5Y cells were seeded in 96-well plates at a density of 10^4^ cells/well and exposed to 0, 5, 10, 15, and 20 μM spinetoram, respectively, for 24 h and 48 h. Then, MTT solutions (5 mg/mL) were added to each well for 4 h. Next, the medium was removed, and the DMSO solution was added to the well. Finally, the absorbance was measured using the microplate readers (BioTeck, Portland, OR, USA) at the wavelength of 570 nm, and then the relative cell viability was calculated.

### 4.4. Mitochondrial Membrane Potential (ΔΨm) Analysis

The mitochondrial membrane potential (ΔΨm) was assessed by measuring the relative fluorescence of rhodamine 123. SH-SY5Y cells were seeded in 60 mm plates and exposed to 0, 5, 10, 15, and 20 μM spinetoram, respectively, for 24 h. After treatment, cells were stained with 1 μg/mL in fresh medium for 30 min at 37 °C. The fluorescence signal was observed and captured using a fluorescence microscopy (Leica, Wetzlar, Germany) at 488 nm excitation.

### 4.5. Measurement of Mitochondrial Permeability Transition Pore (mPTP), Intracellular ATP and Ca^2+^ Level

The mPTP was measured by monitoring the fluorescence of calcein-AM. Briefly, SH-SY5Y cells were seeded in 60 mm plates and exposed to 0, 5, 10, 15, and 20 μM spinetoram, respectively, for 24 h. The cells were superfused with 1 mM of calcein-AM at 37 °C for 20 min, followed by incubation with 1 mM of CoCl_2_ for 30 min. The fluorescence was photographed and examined using fluorescence microscopy. The intracellular ATP level was measured by ATP assay kit (Beyotime, Shanghai, China). After exposure to spinetoram, the cells were harvested, and the protein concentrations were measured using a BCA Protein Assay Kit. Then, the activity of ATP was determined according to the manufacturer’s protocol. The luminescence was measured by a microplate reader (BioTeck, USA). The intracellular Ca^2+^ level was assessed by cell-permeable dye Fluo-3 AM as an indicator. After exposure to spinetoram, the cells were incubated with 2 μM of Fluo-3 AM (Beyotime, China) for 30 min to allow the fluorescent dye to diffuse into the cells. The flow cytometer was used for the detection of fluorescence intensity.

### 4.6. Monodansylcadaverine (MDC) Staining Assay

SH-SY5Y cells were seeded in 60 mm plates and exposed to 0, 5, 10, 15, and 20 μM spinetoram, respectively, for 24 h. After treatment, the cells were incubated with 1 mg/mL of MDC (Beyotime, China) for 15 min. The autophagic vesicles were observed and captured under fluorescence microscopy at the ultraviolet wavelength.

### 4.7. Detection of ROS Content and Enzyme Activity

SH-SY5Y cells were seeded in 60 mm plates and exposed to 0, 5, 10, 15, and 20 μM spinetoram, respectively, for 24 h. The intracellular reactive oxygen species (ROS) level was detected by fluorescent redox probe DCFH-DA. After exposure to spinetoram, the cells were incubated with 10 μM of DCFH-DA (Sigma-Aldrich, USA) for 30 min. Then, the fluorescence intensity was determined by a flow cytometer (Beckman Coulter, Brea, CA, USA). Other cells were harvested, and then the protein concentrations were measured using a BCA Protein Assay Kit (Thermo Scientific, Waltham, MA, USA).

MDA levels were analyzed with the Malondialdehyde (MDA) content assay kit. Samples were processed according to the manufacturer’s instructions and a microplate reader (BioTeck, USA) was used to measure the optical density (OD) value at 532 nm and 600 nm. The activity of antioxidant enzymes (CAT, SOD, GSH-Px) was determined according to their assay kit manufacturer’s protocol. The OD values of the treated samples were also measured using a microplate reader, where CAT activity was detected at 240 nm, SOD activity was detected at 560 nm, and GSH-px activity was detected at 412 nm. The data for these measurements are presented as relative values for the control group.

### 4.8. Western Blot Analysis

SH-SY5Y cells were seeded in 60 mm plates and exposed to 0, 5, 10, 15, and 20 μM spinetoram, respectively, for 24 h. After treatment, the protein concentrations within cells were measured using BCA protein assay kits. The western blotting assay was performed as previously described [36] using LC3, Beclin1, p62, p-mTOR, p-AMPK, and GAPDH antibodies (1:1000), and a secondary antibody (1:4000). The protein signal was detected by an enhanced chemiluminescence system (Tanon, Shanghai, China), and the relative optical density of each band was analyzed using Image J V2.3.0 software.

### 4.9. Zebrafish Husbandry

The wild type (AB) zebrafish was obtained from the China Zebrafish Resource Center (Wuhan). Adult zebrafish were cultured in a separate water circulation system with a light cycle of 14 h/10 h (light/dark) per day at 28 ± 1 °C. Zebrafish were fed with fresh shrimp twice a day. Prior to breeding, the female and male fish were placed in the spawning tank (in a 1:1 or 2:1 ratio) and subjected to a dark treatment. The fertilized eggs were collected shortly after mating. The unfertilized eggs and broken fertilized eggs were eliminated using a stereomicroscope (Z7450T, Nikon, Tokyo, Japan), and the remaining intact fertilized eggs were then incubated in an incubator at 28 °C for the following experiments.

### 4.10. Neuromotor Behavior of Zebrafish

The embryos at 6 hpf (hours post-fertilization) were randomly placed into 6-well plates (20 embryos/well) and treated with 0, 5, 10, and 20 μg/mL of spinetoram, respectively. Approximately 18 hpf embryos were collected and tail swings was measured by manually counting the number of tail wiggles in 20 s. Zebrafish larvae at 120 hpf were individually positioned in 96-well plates, with one tail per well. The plates were subsequently assessed using a stereomicroscope (Semilab, Shanghai, China) in order to examine the spontaneous locomotor rate of the zebrafish larvae under continuous 10-min light exposure.

### 4.11. Immunofluorescence Assay

The zebrafish embryos were randomly placed into 12-well plates (10 embryos/well) and exposed to 0, 5, 10, and 20 μg/mL of spinetoram solution for 120 hpf. After treatment, the immunofluorescence assay was performed according to a published protocol [37] using the LC3 antibody (1:200) and the Alexa fluor 488-conjugated antibody (1:200). The images were obtained with a laser scanning confocal microscope at a wavelength of 488 nm.

### 4.12. Statistical Analysis

The experiment was conducted thrice to ensure accuracy, and the results were analyzed using SPSS 23.0 software. The cell activity assay data were analyzed through two-way ANOVA, while the remaining experiments were analyzed through one-way ANOVA and Tukey’s test to establish statistical differences. Mean ± standard deviation (SD) values are presented in this study. The ‘*’ and ‘**’ indicate significance levels of *p* < 0.05 and *p* < 0.01, respectively.

## Figures and Tables

**Figure 1 molecules-29-00253-f001:**
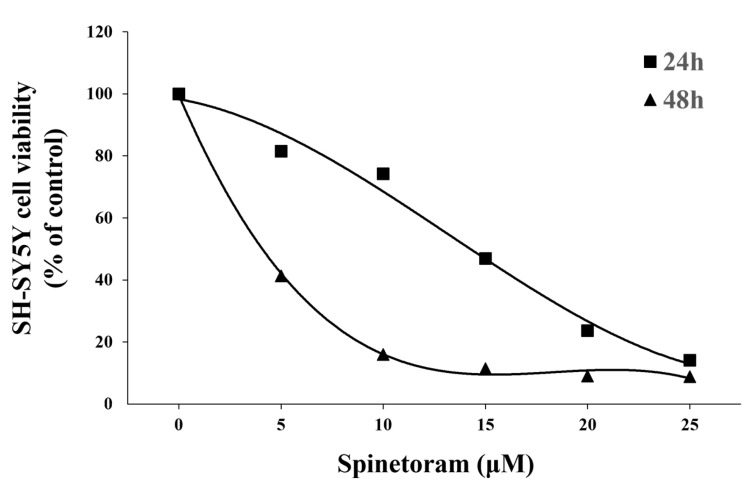
Effect of varying concentrations (0 μM, 5 μM, 10 μM, 15 μM, 20 μM, 25 μM) of Synephrine on the viability of SH-SY5Y cells for 24 h and 48 h.

**Figure 2 molecules-29-00253-f002:**
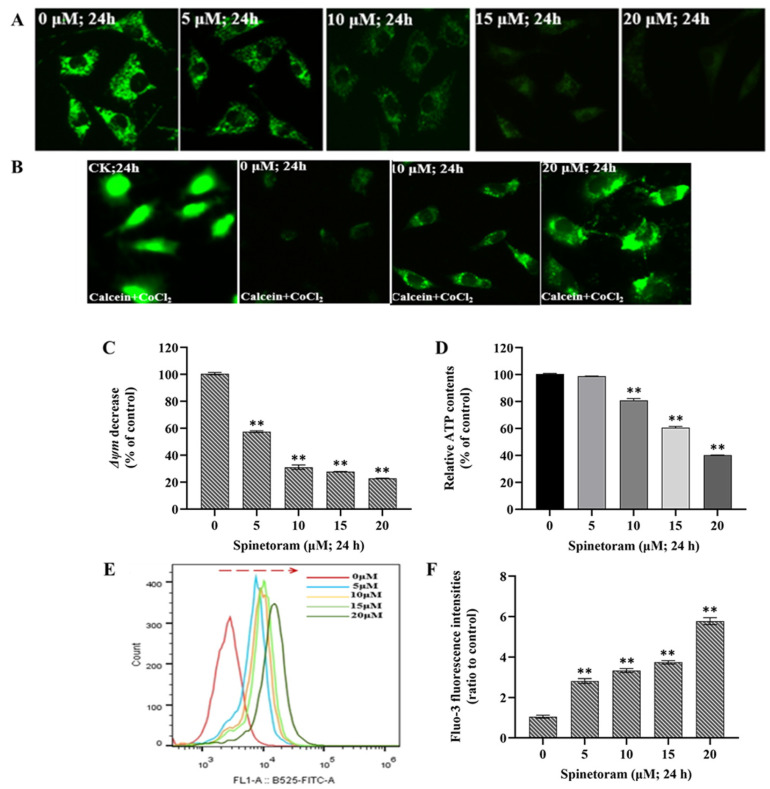
Effects of spinetoram on mitochondrial functions in SH-SY5Y cells. (**A**) The mitochondrial membrane potential (ΔΨm) was detected by Rh-123 probe. (**B**) Mitochondrial permeability transition pore (mPTP) was detected with calcein-AM and CoCl2 dye. (**C**) The data of relative fluorescence intensity of Rh-123 probe. (**D**) The relative ATP level. (**E**,**F**) The relative Ca^2+^ level was detected with Fluo-3 AM probe and the data of relative fluorescence intensity. The experiments were repeated three times and shown as means ± SD. The significance level was presented as ** *p* < 0.01 vs. the control.

**Figure 3 molecules-29-00253-f003:**
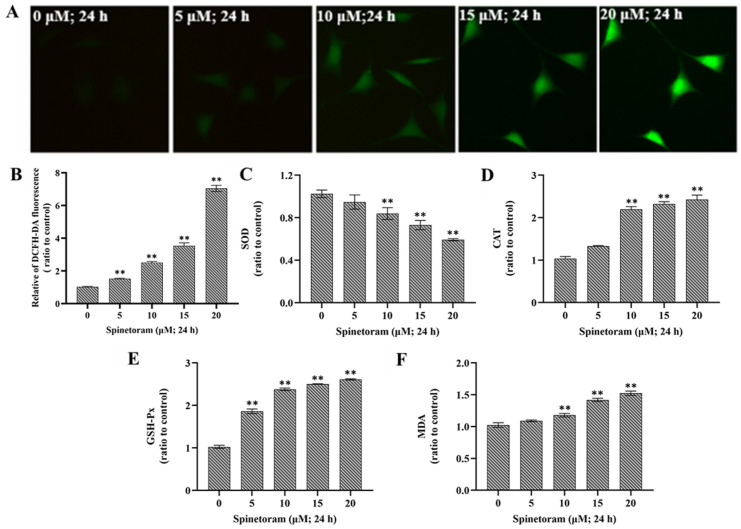
Effect of spinetoram on oxidative damage in SH-SY5Y cells. (**A**,**B**) ROS level was detected with DCFH-DA probe and the data of relative fluorescence intensity. (**C**–**F**) The relative contents of SOD, CAT, GSH-Px, and MDA in cells exposed to spinetoram for 24 h. The experiments were repeated three times and shown as means ± SD. The significance level was presented as ** *p* < 0.01 vs. the control.

**Figure 4 molecules-29-00253-f004:**
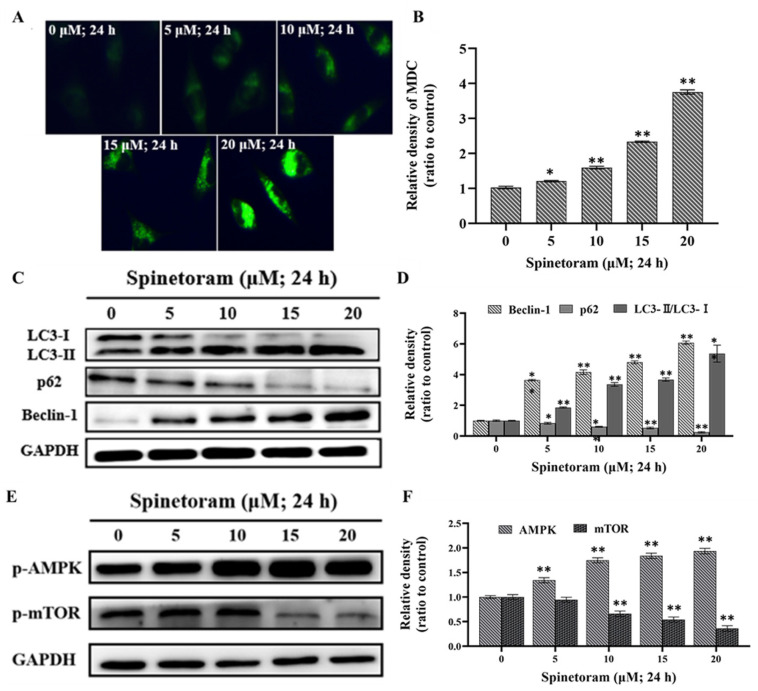
Effect of spinetoram on autophagy in SH-SY5Y cells. (**A**,**B**) The autophagosomes were detected with MDC dye. (**C**,**D**) The expression of LC3-I, LC3-II, p62, and beclin-1 protein detected by western blot assay. (**E**,**F**) The expression of p-AMPK and p-mTOR protein in cells. The experiments were repeated three times and shown as means ± SD. The significance level was presented as * *p* < 0.05, ** *p* < 0.01 vs. the control.

**Figure 5 molecules-29-00253-f005:**
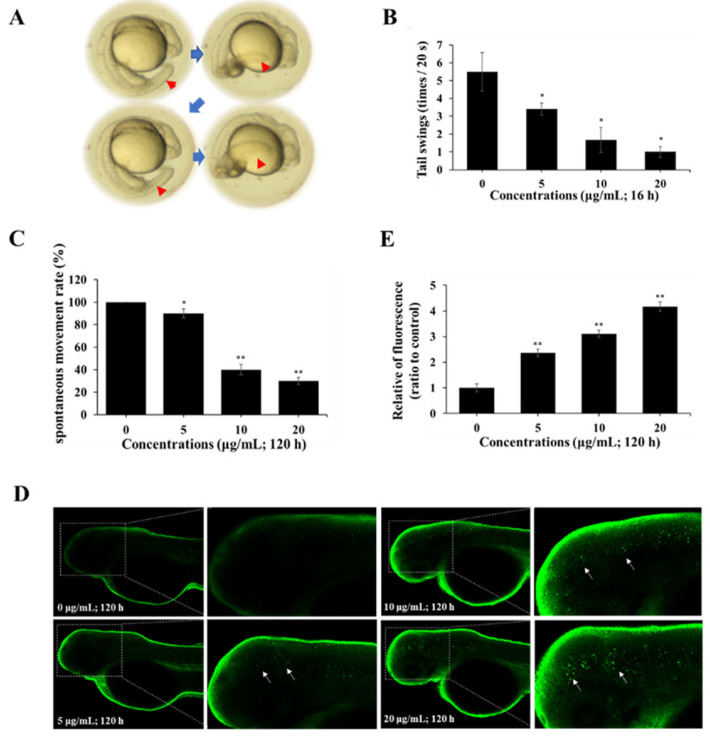
Effect of spinetoram on neurobehavior in zebrafish. (**A**,**B**) The spontaneous movements (tail swings, blue arrows and red arrows point to tail swings) of embryos exposed to spinetoram up to 18 hpf. (**C**) The spontaneous movements of embryos exposed to spinetoram for 120 hpf. (**D**,**E**) The autophagy of embryos exposed to spinetoram for 120 hpf (white arrows point to autophagic cells). The experiments were repeated three times and shown as means ± SD. The significance level was presented as * *p* < 0.05, ** *p* < 0.01 vs. the control.

## Data Availability

The datasets generated or analyzed during this study are available from the corresponding author on reasonable request.

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
