# Peer review of "Spinetoram-Induced Potential Neurotoxicity through Autophagy Mediated by Mitochondrial Damage"

_molecules, 2024, doi:10.3390/molecules29010253_

Round 1
Reviewer 1 Report (Previous Reviewer 2)
Comments and Suggestions for Authors
Dear Editor, many thanks for your confidence for giving me the opportunity to review the manuscript entitled “Spinetoram induced potential neurotoxicity through autophagy 2 mediated by mitochondrial damage” with Manuscript ID: molecules-2725380 for the Molecules Journal.
Authors have replied for most of comments send by reviewer. However, there are still some comments need response from authors and not explained as following
Line 28-29: Conclusion please illustrate what is toxicological data for the safety risk assessment of Spinetoram is provided.
Material and methods section:
- Line 184 what do you mean by hpf full name should be mentioned for the first time. Does author mean hours post fertilization please describe in the text.
- Number of samples examined in each group should be mentioned
- Statistical analysis: here author mentioned that time was 24 or 48 hours and different concentrations were used thus the correct statistical analysis test is two way anova as for time and concentrations (two factors)
Author Response
please see the attachment.

Reviewer 2 Report (Previous Reviewer 3)
Comments and Suggestions for Authors
Although some improements were made, I still have doubts such as:
- Sangon Biotech website is in Chinese making the search for these kits almost impossible. As such, further information needs to be included (e.g wavelengths, probes, reagents, etc).
- There must be something wrong in the behavioural methodology applied. While the authors have corrected the methods stating that “zebrafish larvae at 120 hpf were individually positioned in 96-well plates, with one tail per well”, the figure 5C shows more than one larve in a well that seems bigger than that of a 96-well plate. This raises doubts regarding the methods applied.
- The discussion still lacks comparison of data. For instance, the higher concentration testes in zebrafish correspond to around 27 uM which is similar to thar applied in cells. However, there is no comparison or discussion of the findings between both models. Also, neurobehavioural findings should be cautiously interpreted based on this rudimental analysis which should be complemented by other methods before concluding about the potential behavioural outcomes. In addition, there is no discussion of this findings nor comparison with other published data on other insecticides.
Author Response
Dear reviewer,
Thank you for your interest in our manuscript, and the following are responses to your comments:
- Sangon Biotech website is in Chinese making the search for these kits almost impossible. As such, further information needs to be included (e.g wavelengths, probes, reagents, etc).
As you recommended, we have included more information in the Experimental Methods section utilizing these kits.
- There must be something wrong in the behavioural methodology applied. While the authors have corrected the methods stating that “zebrafish larvae at 120 hpf were individually positioned in 96-well plates, with one tail per well”, the figure 5C shows more than one larve in a well that seems bigger than that of a 96-well plate. This raises doubts regarding the methods applied.
We have identified issues with the image we previously provided, as it does not accurately reflect our experimental methodology. That's why we removed it, but it doesn't impact the presentation of our experimental results.
- The discussion still lacks comparison of data. For instance, the higher concentration testes in zebrafish correspond to around 27 uM which is similar to thar applied in cells. However, there is no comparison or discussion of the findings between both models. Also, neurobehavioural findings should be cautiously interpreted based on this rudimental analysis which should be complemented by other methods before concluding about the potential behavioural outcomes. In addition, there is no discussion of this findings nor comparison with other published data on other insecticides.
Thanks to your suggestions, we have revised and added to the discussion section.

Round 2
Reviewer 2 Report (Previous Reviewer 3)
Comments and Suggestions for Authors
No more comments to the authors.
This manuscript is a resubmission of an earlier submission. The following is a list of the peer review reports and author responses from that submission.
Round 1
Reviewer 1 Report
Comments and Suggestions for Authors
In the paper entitled "Spinetoram induced potential neurotoxicity through autophagy mediated by mitochondrial damage" Contains some intersting findings in vitro and in vivo in zebrafiah, hoever in its present state there are number of issues that need to be addressed.
1. Figure 1: a does-response experiment like the one described should not be represented as a histogram, it is impossible to calculate an LD50 from it. The correct representation should be nonsigmoidal regression line with variable slope.
2. IC50 is defined as conentration in which the test compound inhibits half of the reaction. This is NOT what figure 1 shows or calculates. The correct term is LD50, dose at which 50% of the population is dead. You are looking at viability not inhibition of an enzymatic process.
3. SH-SY5Y cells are a dopaminergic cell line and there is no mention of that fact or assays that could measure dopaminergic functions in vitro.
4.In figure 3, the legend describes SOD, catalase and GSH Px contents of the cell and presents the results as histograms. The resutls section describes measuring enzymatic activity for these enzymes. Which is it? The method section does not describe the kits at all. Were these ELISA kits measuring levels of these proteins or were these assays to measure the enzyme's activity. These need to be defined. Am I looking at a histogram of a quantitative ELISA or the activity of the enzyme? Or am I looking at a densitometric analysis from western blots? I can't tell, therefore I can not trust the experiment was done correctly.
5. Figure 5:The behavioral analysis of zebrafish by spontaneous movements is very broad and doesn't reflect just neuronal functions. There is literature that use this same assay to reflect development of muscles. A more specific behavioral assay should be selected.
6. MInor comment, but MMP is not the standard abbreviation for mitochondrial membrane potential, should be Dym
Comments on the Quality of English Languagenone to report
Reviewer 2 Report
Comments and Suggestions for Authors
Dear Editor, many thanks for your confidence for giving me the opportunity to review
the manuscript entitled “Spinetoram induced potential neurotoxicity through
autophagy 2 mediated by mitochondrial damage” with Manuscript ID:
molecules-2638899 for the Molecules Journal.
All comments are cited in the attached reviewed manuscript.
1- Abstract, unfortunately can't stand alone, main aim of the study is not
illustrated. Several points should be clarified for more illustration such as dose
level of Spinetoram either in cell line or in vivo study in zebra fish.
Neurobehavioral abnormalities. Significance of the obtained results
2- Conclusion please illustrate what is toxicological data for the safety risk
assessment of Spinetoram is provided.
3- Introduction: Need more reviewing data
Line 39 -40: author mentioned properties of polymyxin as insecticide but it is
not the insecticide of the study what about Spinetoram????
Generally, neurotoxic effects of the Spinetoram and its significance should be
more illustrated and reviewed in the section of introduction. As well what is the
main problem the current study aimed to investigate, in vivo and in vitro studies
Material and methods section:
- Kits product number should be illustrated for all the used kits
- Line 105 what was the final concentration of the Spinetoram,
additionally illustrate number of experimental groups and exposure
levels
- Line 164-169 the test of neuromotor activity should be mentioned in
more detailed manner regarding description and time
- Line 166 what do you mean by hpf full name should be mentioned for
the first time
2
- Illustrate why these concentrations of the Spinetoram selected for the
study what is its relation to the LD50 or citing reference for the selected
dose in relation to the doses of human exposure.
- -
For oxidative stress related indices why lipid peroxidation only
measured what about total antioxidant capacity, and specific marker of
oxidative DNA damage (8-hydroxy-2-deoxyguanosine)
- Number of samples examined in each group should be mentioned
- Statistical analysis:
- Line 109-110: here author mentioned that time was 24 or 48 hours
and different concentrations were used thus the correct statistical
analysis test is two way anova as for time and concentrations (two
factors)
- The post hoc test Tukey's post hoc is more accurate than Duncan
test, so statistical significance should be reanalyzed again using Tukey
test.
4- Results:
- figure 1 ligand should illustrate the dose levels of Spinetoram concentrations
The effects of various concentrations of Spinetoram on the viability of SH-SY5Y
cells were mentioned for 24h or 48 hr, which concentration was more significant.
5- Discussion section:
- After carefully reinvestigating their preliminary observations, the author
could improve the quality of discussion better by integrating the results with
the current literature for each discussed parameter.
Reviewer 3 Report
Comments and Suggestions for Authors
The authors present a study dedicated to the neurotoxicological evaluation of spinetoram by assessing cell viability, mitochondrial alterations, autophagy and complemented the study with zebrafish assays. Although presenting novel data, some issues need to be solved:
- L22, which neurobehavioral abnormalities were observed in zebrafish?
- L52, this compound has been previously tested in HepG2 cells (doi: 10.1080/09540105.2019.1650900) by the same authors. This sentence should be revised.
- L109, why were these concentrations tested?
- L166, how do these concentrations relate with those tested in cells? Why were these concentrations tested?
- L165, embryos of which age? How were embryos obtained?
- L168, how was the spontaneous movement assessed at 120 hpf?
- L171, at which time-point did the exposure begun?
- L179, the Duncan’s test is not recommended due to inadequate error rate adjustment (DOI 10.7717/peerj.10387). As such, statistics need to be revised using other post-hoc comparion method.
- L192, remove references from the results section. Review along the manuscript.
- L216, oxidative stress can be measured by several other biomarkers.
- L223, error bars are missing in the control group. Review throughout the manuscript.
- L255, spinetoram was exposed up to 120 hpf, not “for 120 hpf”.
- L267, tail coiling begins at 17 hpf and peak at 19 hpf (doi: 10.3390/w13020119, 10.1016/j.ecoenv.2019.109754). How can the authors measure them at 16 hpf? In addition, how can the measurement of zebrafish be analysed by visual inspection (material and methods sections is poorly described) if a normal zebrafish swims around 1-3 mm/seg being out of inspection area in this time?
- The overall discussion is too poor without the proper interpretation of the findings. For instance, there is no comparison of the data obtained with available literature for both cell lines and zebrafish.